# Cine-MRI detection of abdominal adhesions with spatio-temporal deep learning

**Bram de Wilde**[1]                                         BRAM.DEWILDE@RADBOUDUMC.NL

**Richard P. G. ten Broek**[2]                              RICHARD.TENBROEK@RADBOUDUMC.NL

**Henkjan Huisman**[1]                                      HENKJAN.HUISMAN@RADBOUDUMC.NL

[1] *Diagnostic Image Analysis Group, Radboud University Medical Center, Nijmegen, The Netherlands*

[2] *Department of Surgery, Radboud University Medical Center, Nijmegen, The Netherlands*

## Abstract

Adhesions are an important cause of chronic pain following abdominal surgery. Recent developments in abdominal cine-MRI have enabled the non-invasive diagnosis of adhesions. Adhesions are identified on cine-MRI by the absence of sliding motion during movement. Diagnosis and mapping of adhesions improves the management of patients with pain. Detection of abdominal adhesions on cine-MRI is challenging from both a radiological and deep learning perspective. We focus on classifying presence or absence of adhesions in sagittal abdominal cine-MRI series. We experimented with spatio-temporal deep learning architectures centered around a ConvGRU architecture. A hybrid architecture comprising a ResNet followed by a ConvGRU model allows to classify a whole time-series. Compared to a stand-alone ResNet with a two time-point (inspiration/expiration) input, we show an increase in classification performance (AUROC) from 0.74 to 0.83 ($p < 0.05$). Our full temporal classification approach adds only a small amount (5%) of parameters to the entire architecture, which may be useful for other medical imaging problems with a temporal dimension.

**Keywords:** cine-MRI, adhesion detection, spatio-temporal, ConvGRU

## 1. Introduction

Up to 20% of patients undergoing abdominal surgery develop chronic pain due to post-operative adhesions (van der Wal et al., 2011). Recently, cine-MRI has been introduced as an effective method to detect these adhesions non-invasively (Lang et al., 2008). Non-invasive detection plays a key role in patient management, as it prevents both unnecessary surgeries and severe complications during surgery (van den Beukel et al., 2018). Radiological interpretation of cine-MRI, however, is time-consuming and strongly depends on expertise. Adhesions are very thin tissue structures, which in itself are invisible on a single time frame on MRI or other imaging modalities. During the scan, patients are instructed to perform the Valsalva maneuver repeatedly, thereby inducing motion in the entire abdomen. The radiologist detects an adhesion by its property to connect different structures, appearing as a local absence of sliding motion on the entire cine-MRI time-series.

In this work, we approach adhesion detection as a classification problem and efficiently model spatio-temporally using a hybrid architecture. A feed-forward CNN (ResNet-18) extracts low dimensional spatial features. These feature maps allow temporal aggregation with a lightweight recurrent neural network, ConvGRU, which models spatial information through time. We show that this approach works for adhesion detection and expect that it applies equally well to any medical imaging task with a temporal dimension.

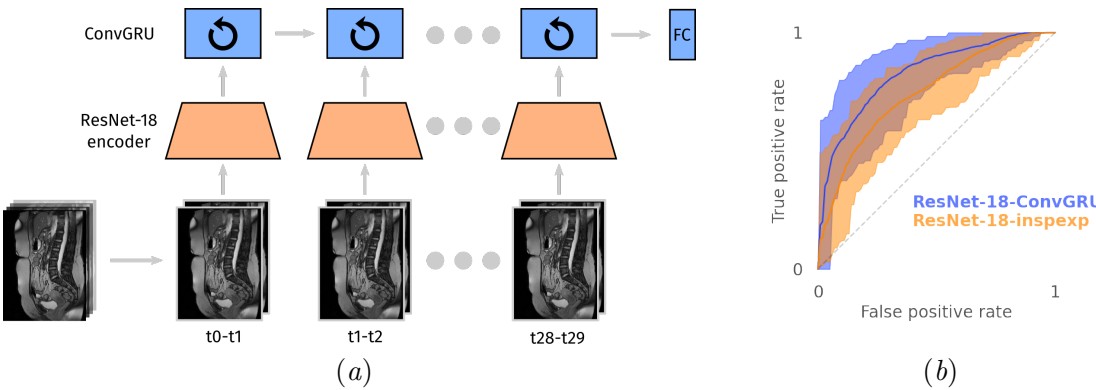

Figure 1: (a) A schematic overview of Resnet-18-ConvGRU. Each consecutive frame pair is fed to a ConvGRU model, through a ResNet-18 encoder. A final fully connected layer outputs a probability score. (b) ROC curves of both models, with 95% confidence intervals estimated with bootstrapping.

## 2. Methods

All cine-MRI series used in this work are sagittal abdominal series acquired at a single center in the Netherlands and were annotated by an experienced radiologist. Patients were scanned because of clinical suspicion of adhesions. The total number of series is 104, taken from the scans of 63 patients. Each series has a dimensionality of $30 \times 256 \times 192$ ($T \times H \times W$), with a time between each frame of 0.4 seconds.

The baseline architecture, referred to as ResNet-18-inspexp, is a ResNet-18 that receives a single frame pair (two time points) as 2-channel input (He et al., 2016). These frames are pre-selected such that the difference in abdominal position is largest. In this model, temporal information is used by choosing the two most relevant time points of the series. We also experimented with taking consecutive frame pairs, but that was inferior to the method described above. These results are excluded here for brevity.

The proposed architecture (ResNet-18-ConvGRU), draws inspiration from recent work on video processing with recurrent networks, using a GRU-based architecture with convolutional instead of fully connected layers (Ballas et al., 2016; Zhu et al., 2019). The ResNet-18-inspexp model is used as a pre-trained encoder, stripping away its fully connected layers. The resulting low-dimensional activation maps are fed to a ConvGRU model, a recurrent neural network that can efficiently model spatio-temporal data, as in illustrated in Figure 1(a). This allows ResNet-18-ConvGRU to model full temporal data as opposed to the two time-point ResNet baseline.

Model performance is evaluated using 5-fold cross validation, with unique patients in each fold. The performance metric AUROC is obtained over the full dataset, by aggregating the validation predictions of each fold. 95% confidence intervals are estimated with bootstrapping. P-values for the difference in AUROC between models are estimated using a permutation test.

## 3. Results

ResNet-18-ConvGRU performs significantly better ($p = 0.002$) than ResNet-18-inspexp (see Figure 1($b$)) with an AUROC (95%CI) of 0.83 (0.70-0.93), as opposed to 0.74 (0.60-0.87).

## 4. Discussion

A lightweight architecture based on a ConvGRU model outperforms a handcrafted two time-point temporal classification approach. By adding only about 5% of the weights of the baseline classifier, it can exploit the full temporal dimension as shown by a substantial increase in performance. Based on a currently running observer study (results to be published), the observed model performance (AUROC 0.83) compares to a radiologist with moderate experience. The method seems promising to aid radiologists with detection of adhesions on cine-MRI.

Other work using ConvGRU in a similar manner, e.g. (Zhu et al., 2019), typically uses high resolution video input and uses larger 3D models as encoders. We show that a small ResNet-18 encoder suffices in the case of low resolution input, possibly allowing for computationally tractable end-to-end learning. With enough compute, this approach may also be a viable way to process high-resolution 4D medical data, using 3D encoders. Generating saliency maps may give more insight into the adhesion localization capacity of the model. It may also be possible to convert the model to a detection model, by regressing the ConvGRU output on bounding box parameters instead of a binary label.

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
