# OpenReview forum: "Cine-MRI detection of abdominal adhesions with spatio-temporal deep learning"
_MIDL.io/2021/Conference/Short — MIDL 2021 Poster_

### Official Review · Reviewer_KidC · 2021-04-22

**Confidence:** 4
**Final Rating:** 4

**Summary:**

The paper describes a method for detecting abdominal adhesions in cine-MRI by combining a convolutional encoder with a lightweight recurrent network. The method shows promising improvement over a ResNet baseline. Citing an ongoing observer study, the achieved performance is claimed to be similar to a moderately experienced radiologist.

**Strengths:**

The introduction is clear, both in clarifying the clinical use for the method, as well as in motivating the need for aggregating temporal information.

The description of the method is very clear and easy to follow. While combining frame-by-frame convolutional encoders with a recurrent neural network in cine-MRI is not new (e.g. for cardiac segmentation, by Poudel et al., 2016), it is clearly a sensible method for this specific application.

The baseline seems like a reasonably strong approach, yet the presented method outperforms it by a large margin.

**Weaknesses:**

The method is compared to a baseline that requires more than thirty times less computational power. In terms of practical applicability, it could be considered more "fair" to compare to an ensemble of 30 ResNet predictions based on randomly selected time point combinations.

The method is a diagnostic tool, but it offers no interpretability of its results (such as a shear map or a localization of the adhesions). Hence, it is not usable in clinical practice. The authors identify such options as future work, meaning this method could be considered a stepping stone towards a system that _is_ interpretable.

**Deanonymize Review:**

yes

**Detailed Comments:**

The text in Fig 1b is too small to read on a printed copy of the paper. Consider increasing the font size.

Percentages are denoted in different styles ("20 percent" in the introduction vs "5%" in the discussion). Either is fine, but keeping the style consistent would be preferable.



**Justification Of The Rating:**

This work presents what seems to be the first fully automatic method for detecting abdominal adhesions in cine-MRI. The paper is well-written and the presented method is explained clearly. Furthermore, the the validation method is reasonably strong and the achieved results are promising.

**Paper Type:**

both

**Special Issue:**

no

---

### Official Review · Reviewer_dXxk · 2021-04-30

**Confidence:** 4
**Final Rating:** 4

**Summary:**

The authors proposed a spatiotemporal deep learning approach, inspired by convGRU for noninvasive detection of abdominal adhesions from cine MRI. The focus was to classify presence or absence of adhesions in sagittal abdominal cine-MRI series
The method is validated on a dataset of 104 cine MRI scans coming from 63 patients.

**Strengths:**

1.	The Clinical motivation of the paper is clear. I liked the way the authors detailed the clinical problem and motivation.
2.	The paper is well-written and easy to follow.
3.	The method and results make sense.

**Weaknesses:**

I don’t have any major criticism about this short paper. Some minor concerns:
1.	The axes of Figure 1b is illegible. Please make sure that the readers can understand the content in final version.
2.	If possible, I’d urge the authors to add some zoom in pictures of the cohesion as this is a very new diagnostic imaging technique of which most are not aware of.


**Deanonymize Review:**

no

**Justification Of The Rating:**

As I already mentioned, I am quite satisfied by the contribution of the authors. But before acceptance I’d like the authors to address the issues mentioned in weakness.

Changes in Final Version
Please see the weaknesses section.


**Paper Type:**

validation/application paper

**Special Issue:**

no

---

### Meta-Review · Area_Chair_wqHc · 2021-05-09

**Recommendation:** Accept (Poster)
**Confidence:** 5

**Metareview:**

Both reviewers give strong support for the paper with clear clinical motivation. I recommend acceptance and encourage the authors to take into account the minor suggestions for further improvements and requesting the approval for source code release.

---

### Decision · Program_Chairs · 2021-05-11

Accept (Poster)